# Train–Track–Bridge Dynamic Interaction on a Bowstring-Arch Railway Bridge: Advanced Modeling and Experimental Validation

**DOI:** 10.3390/s23010171

**Published:** 2022-12-24

**Authors:** Diogo Ribeiro, Rui Calçada, Maik Brehm, Volkmar Zabel

**Affiliations:** 1CONSTRUCT-LESE, School of Engineering, Polytechnic of Porto, 4249-015 Porto, Portugal; 2CONSTRUCT-LESE, Faculty of Engineering, University of Porto, 4200-465 Porto, Portugal; 3Merkle CAE Solutions GmbH, 89518 Heidenheim, Germany; 4Institute of Structural Mechanics, Bauhaus-University Weimar, 99423 Weimar, Germany

**Keywords:** train–track–bridge interaction, railway bridge, advanced numerical modeling, dynamic testing, validation

## Abstract

This article describes the validation of a 3D dynamic interaction model of the train–track–bridge system on a bowstring-arch railway bridge based on experimental tests. The train, track, and bridge subsystems were modeled on the basis of large-scale and highly complex finite elements models previously calibrated on the basis of experimental modal parameters. The train–bridge dynamic interaction problem, in the vertical direction, was efficiently solved using a dedicated computational application (TBI software). This software resorts to an uncoupled methodology that considers the two subsystems, bridge and train, as two independent structures and uses an iterative procedure to guarantee the compatibility of the forces and displacements at the contact points at each timestep. The bridge subsystem is solved by the mode superposition method, while the train subsystem is solved by a direct integration method. The track irregularities were included in the dynamic problem based on real measurements performed by a track inspection vehicle. A dynamic test under traffic actions allowed measuring the responses in the bridge, track, and vehicles, which were synchronized by GPS systems. The test results demonstrated the occurrence of upward displacements on the deck, which is a characteristic of structures with an arch structural behavior, as well as an alternation of tensile/compressive stresses between the rail and deck due to the deck–track composite effect. Furthermore, the acceleration response of the bridge proved to be significantly influenced by the train operating speed. The validation procedure involved comparing the dynamic responses obtained from the train–bridge interaction model, including track irregularities, and the responses obtained experimentally, through the test under traffic actions. A very good correlation was obtained between numerical and experimental results in terms of accelerations, displacements, and strains. The contributions derived from the parametric excitation of the train, the global/local dynamic behavior of the bridge, and the excitation derived from the track irregularities were decisive to accurately reproduce the complex behavior of the train–track–bridge system.

## 1. Introduction

Railway bridges are structures in which the dynamic effects induced by traffic can reach significant values. These effects are currently of rising importance due to the increase in the train’s operating speed, axle loads, and modifications in the number of axles and axle spacing. Additionally, the structural typologies of the bridges, using increasingly light and resistant materials, create new demands from the point of view of the dynamic behavior of the train–track–bridge system, especially in terms of structural stability, running stability, and passenger comfort [1].

The dynamic behavior of the train–track–bridge system is highly dependent on the dynamic interaction among the bridge, track, and vehicle subsystems [2,3,4,5]. For this purpose, advanced numerical models have been developed for the train, track, and bridge subsystems, including the wheel–rail and track–deck interfaces, as well as the track irregularities. The accuracy of these models is strongly dependent on the validation of the numerical results based on dedicated dynamic tests under railway traffic [6,7,8,9,10,11].

Regarding the bridges, they are usually modeled through the finite element (FE) method using three-dimensional (3D) elements to model both the structure and the track [2,3]. These models can accurately reproduce the behavior of the track–bridge coupling system, independently of its complexity, since the several components of the bridge (deck, bearing supports, piers, and foundations) and track (ballast, sleepers, pads, fasteners, and rails) may be modeled using different types of FEs (beam, shell, or solid) and connecting elements (rigid links and spring–dashpot assemblies). The inclusion of the ballasted track in the numerical models of the bridges has several advantages, since this approach (i) guarantees the adequate train loads distribution, (ii) filters high-frequency content from the bridge’s dynamic response, (iii) incorporates the track–bridge composite effect due to the longitudinal shear stress transmission occurring between rails and bridge deck, through the ballast layer, and (iv) simulates the track continuity between neighboring decks and between the deck and the embankment [10,11,12,13].

Regarding the modeling of railway vehicles, in most situations, it is based on multibody dynamic formulations [4,14,15] or numerical models relying on the FE method [16,17,18]. In the formulations based on multibody dynamics, the carbody, bogies, and axles of the vehicles are modeled by means of rigid bodies connected by springs and dampers to simulate the primary and secondary suspensions. Otherwise, in formulations based on the FE method, it is possible to consider the deformability of the carbody, bogies, and axles. Several authors such as Carlbom [19] and Diana et al. [20] showed that the flexural modes of the carbody can have an important contribution to the accelerations to which the passengers are subjected. This effect can be enhanced due to the eventual coupling with the under-chassis equipment. In the works carried out by Carlbom [19] and Wei and Griffin [21], the vehicle models also included the passenger-seat system through simplified models of one or two degrees of freedom. These models allow directly evaluating the acceleration levels on the passengers.

The train–bridge dynamic interaction problem can be solved using two main distinct approaches: (i) a coupled approach, which considers the train–bridge as a coupled system in which the equations of each subsystem are assembled into a global system of equations solved simultaneously [10,22,23]; (ii) an uncoupled approach, where the train and the bridge subsystems are modeled separately and solved using an iterative procedure that guarantees the compatibility of forces and displacements at the contact points within each time-step [11,14]. In both approaches, the dynamic interaction can either be performed solely in vertical direction [10,22] or also include lateral and longitudinal directions [4,24]. 

Within a train–bridge dynamic interaction problem, the wheel–rail contact problem can be solved using distinct strategies, i.e., using simplified methodologies [25], which assume the rigid connection between the wheel and the rail, or using the contact theory [26,27], which admits the existence of relative movements between the wheel and the rail. In the second approach, the wheel–rail contact is generally described by the nonlinear Hertz model [28], for contact in the normal direction, and the Kalker model [26], for contact in the lateral direction. Within the dynamic calculation of the train–bridge system, the Hertzian contact stiffness is typically linearized, resulting in a linear relationship between the contact force and the relative wheel–rail displacement.

Another relevant aspect for the realistic simulation of the dynamic train–bridge problem is the inclusion of track irregularities. Irregularity profiles can be obtained on the basis of a direct measurement of the track geometry using track inspection vehicles [5,29]. Alternatively, a random generation of track irregularity profiles, based on power spectral density functions, has been adopted by several authors [3,24,30]. These functions have been proposed by several railway administrations based on a wide range of experimental measurements of irregularities profiles; therefore, their validity is restricted to specific wavelength ranges.

The validation of numerical models of the train–bridge system is generally based on dynamic measurements derived from forced vibration tests under traffic actions [31,32,33,34,35,36,37,38,39]. Authors such as Zhai et al. [12], Kwark et al. [14], Liu et al. [2], and Chellini et al. [40] compared the numerical responses, derived from advanced train–bridge dynamic interaction models, with experimental responses, and they obtained a very good agreement between both records. Most of the studies mainly focused on bridges belonging to high-speed lines, while a by far smaller number of publications envisaged bridges on conventional lines. Concerning the existing railways bridges, Li and Wu [41], Li et al. [42], and Horas et al. [43] validated highly accurate multiscale FE models using sub-modeling approaches for the fatigue analysis of critical details. In addition, several authors, such as Ticona Melo et al. [11], Malveiro et al. [6,8], and Saramago et al. [10], highlighted some relevant nonlinear incursions of the track–bridge system during a train passage, particularly at the level of the support bearings and track–deck interface. Typically, the validation is performed in terms of the dynamic response of the bridge (displacements, accelerations, and strains), whereas there are still only few studies that included the dynamic responses of vehicles. Among the few works identified, it is important to mention those by Zhai et al. [12] and Xia et al. [44], where the validation of the vehicle’s response was based only on the maximum values of response, while more expressive meaningful presentations of the results allowing their assessment in time or frequency domain were not given. Bragança et al. [45] performed the validation of a numerical model of a freight wagon based on a dynamic test under real operation conditions. The results showed an excellent agreement between experimental and numerical time histories and corresponding frequency spectra related to the responses on both the vehicle platform and the axles. Despite this, the study was only focused on the plain track and did not include bridges.

In this article, the experimental validation of a train–bridge dynamic interaction model of the São Lourenço bridge was carried out. For this purpose, 3D numerical models were developed for the bridge and for the Alfa Pendular train. The numerical model of the bridge included the track, while the numerical model of the train considered the flexibility of the carbody, bogies, and axles. The track irregularity profiles were obtained from measurements performed by a track inspection vehicle. The dynamic analyses of the train–bridge system were based on a dedicated computational tool that considers the two subsystems, bridge and train, independently modeled, and considers their dynamic interaction, in the vertical direction through an uncoupled methodology. The validation of the train–bridge system involved the comparison of the dynamic responses of the bridge and train numerically obtained with the responses obtained in an experimental test under railway traffic.

The present work contributes with some innovative aspects to the existing bibliography:-The validation of the dynamic model of the train–bridge system is based on highly complex and calibrated numerical models of train, track, and bridge subsystems. In addition, the dynamic problem considers real track irregularities measured using a track inspection vehicle. Most of the existing studies did not consider calibrated numerical models [31,32,33], particularly in the case of the track and vehicles, and they normally resorted to calibrated models of lesser complexity [32]. Furthermore, in most cases, the irregularities of the track were randomly generated on the basis of dedicated power spectral density functions proposed by railway administrations [24,30].-Additionally, the validation procedure includes a wide range of measurements performed on the bridge, including primary and secondary elements, as well as on the vehicle and track. Furthermore, the types of measurements are quite extended, including accelerations, displacements, and strains. In the bibliography, few works focused on the simultaneous validation of the dynamic response in the bridge, track, and vehicle subsystems. Even in situations where the response of the bridge and train were evaluated, the number and type of measurements were limited [2,12,14,40].-The dynamic test of the train–track–bridge system used an integrated dynamic monitoring system that guarantees the synchronization of the measurements on the bridge and vehicles by means of precision GPS systems. This level of detail in terms of time accuracy is normally not reported in the bibliography.-The damping coefficient estimates were obtained by applying the logarithmic decrement method to the experimental records in free vibration resulting after the train passage. In most of the existing literature, the estimates of damping coefficients were based on ambient responses obtained for vibration levels significantly lower than those verified under traffic.

## 2. São Lourenço Railway Bridge

### 2.1. Description

The São Lourenço bridge is located at km +158,662 of the Northern line of the Portuguese railways, which establishes the rail connection between the cities of Lisbon and Porto. The bridge’s structure consists of two half-decks with a span of 38 m that support each of the railway traffic lanes. Each half-deck consists of a prestressed concrete slab, with 0.40 m thickness, laterally suspended by two metallic arches. The suspension of the slab is carried out by means of metallic hangers and diagonals close to the start of the arches. Figure 1 shows a lateral view of the São Lourenço bridge and a cross-section of a half-deck in a section close to the start of the arches.

The deck is supported at each abutment by two pot bearings. The distance between the supports is 38.4 m, and the extremities of the deck slab are cantilevers with 1.8 m length. Each half-deck cross-section, with a total width of 7.35 m, consists of a concrete slab laterally supported by two main girders, forming a U-section, and a side footway.

### 2.2. Numerical Modeling

The numerical model of the São Lourenço bridge was created as a three-dimensional finite element model, including the track, developed in the ANSYS software version 12.0 [46]. Figure 2 shows a perspective of the numerical model of the bridge, including a detailed view of the model in the vicinity of one of the supports.

Concerning the bridge, the deck was modeled by means of solid finite elements, while, for the arches, hangers, diagonals, and bracings beam elements were used. The pot bearings were modeled by means of spring–dashpot assemblies. The additional masses associated with coatings, handrail, and connections (plates, bolts, etc.) between elements of the arches and between the elements of the arches and deck were modeled by mass finite elements. Regarding the connection between the ends of the arches and the deck, a monolithic connection was guaranteed by extending the beam elements of the arch inside the solid elements of the deck support block. This procedure guarantees the continuity of the rotations in these connections since the solid FE of the support blocks did not have rotational degrees of freedom. A similar procedure was implemented for the connections between the hangers/diagonals with the deck.

Concerning the track, the rails were represented by beam elements positioned at its center of gravity, while the sleepers, rail pads, and ballast layer were described using solid finite elements. In particular, the sleepers were modeled with a geometry close to the real one, i.e., with a length of 2.60 m and a trapezoidal cross-section with a lower width of 0.30 m, an upper width of 0.15 m, and a height equal to 0.20 m. The rail pads were modeled using a single elastic parallelepiped element with dimensions of 0.15 m in the longitudinal direction, 0.30 m in the transversal direction, and an equivalent height of 0.02 m. The ballast layer presents a height equal to 0.45 m, measured from the deck to the sleeper’s base. Each ballast finite element has a cubic shape with side equal to 0.30 m. The additional ballast height of 0.17 m, counted from the base of the sleepers and validated by an onsite geometric survey, was considered by means of mass elements. An extension of the track corresponding to the length of the deck and about 10 m to the side of each abutment was also modeled to simulate the track over the adjacent embankments.

The structure was discretized into 16,979 solid elements and 1107 beam elements, in a total of 26,754 nodes and 80,029 degrees of freedom.

The bridge’s numerical model was calibrated using an iterative methodology based on a genetic algorithm and resorting on experimental modal parameters, according to the details presented in [47,48]. The modal parameters are associated with 12 global vibration modes, involving global movements of the deck and arches, and 12 local vibration modes, involving vibrations of the hangers and diagonals, and without significant movements of the deck or arches [47]. Previous studies [47,48] also detailed the results of the sensitivity analysis and optimization; therefore, for simplicity and avoid repetition of information, those aspects are restricted to essential points. Table 1 presents the values of the main geometric and mechanical parameters of the numerical model of the bridge. Most of the parameter values refer to the optimal values obtained on the basis of the calibration process.

Figure 3 shows a comparison of the main global modal configurations of the bridge obtained experimentally and numerically. The configurations refer to transversal bending modes of the arches (1, 4, and 8) and bending (2, 3, 5, 9, and 11) and torsion modes of the deck (6, 7, 10, and 12). To simplify the graphical representation, only the points belonging to the deck are presented. In the same figure, the values of the numerical (f_num_) and experimental (f_exp_) natural frequencies and the values of the parameter MAC (modal assurance criterion), indicating the degree of agreement between numerical and identified mode shapes, are also listed. The results showed a very good agreement between numerical and experimental modal parameters. The average error of the natural frequencies was 1.88%, and the average value of the MAC parameter was equal to 0.908.

### 2.3. Track Irregularities

The information about track irregularities was obtained through measurements by the EM 120 track inspection vehicle from the Portuguese infrastructure manager IP (Figure 4). This vehicle measures the longitudinal leveling of the track as a function of the measurement of the distance from the head of the rail, using a laser system installed on the bogie.

Figure 5 shows the longitudinal leveling profiles of the left and right rails, on a section of track between km +158,600 and +159,200, which includes the São Lourenço bridge. These records consider contributions referring to wavelengths between 3 m and 70 m.

The irregularity profiles measured on both rails were practically coincident. In most of the length of the track section, the amplitude of irregularities on both rails did not exceed 5 mm. The maximum amplitude of irregularities was around 12 mm and occurred close to the midspan of the bridge. The measured irregularities, either in terms of peak value or in terms of standard deviation, were within the limits indicated in the standard EN 13848-5 [49].

Figure 6 shows the amplitudes of the power spectra of the irregularities in both rails as a function of the wavenumber (1/*λ*), considering a track length at the approach to the bridge equal to 500 m. The results show that the highest amplitudes were recorded for wavelengths between 40 m and 60 m.

## 3. Alfa Pendular Train

### 3.1. Description

The CPA 4000 series train, known as Alfa Pendular, consists of six vehicles: four motor (BAS, BBS, BBN and BAN) and two hauled (RNB and RNH). The train has a total length of 158.9 m and can reach a speed of 220 km/h. The total weight, for the normal load situation, is 323.3 t. Axle loads range from 128.8 kN to 136.6 kN [18]. Figure 7a shows a perspective of the Alfa Pendular train with the identification of all vehicles. The train loading scheme is presented in [18].

The carbody is formed by a tubular structure in aluminum alloy, made up of welded honeycomb-type panels. The main structure of the bogie, called a chassis, is made of mild steel and consists of two stringers that are connected by means of two tubular beams, forming a double H structure. The bogie chassis is supported on two axles by means of primary suspension springs and dampers. Figure 7b shows a cross-section of the carbody with the identification of its main constituents: base, side walls, and cover.

Figure 8 shows the graph of the dynamic signature of the Alfa Pendular train, as a function of the excitation frequency, for operating speeds of 150 km/h (AP150) and 181 km/h (AP181). These two speeds were the running speeds recorded in the dynamic test under traffic actions (Section 5). The speed of 150 km/h was recorded during maintenance works on the track, close to the bridge. The dynamic signature of a train translates the dynamic excitation of the train and depends only on its geometric characteristics, in particular axle loads and distances between axles, which can be obtained as follows:(1)$$S_{0}\left(f\right)=\underset{i=1,N-1}{\mathrm{MAX}}\sqrt{{\left[\overset{i}{\underset{k=0}{\sum }}P_{k}\cos\left(\frac{2\pi x_{k}}{v/f}\right)\right]}^{2}+{\left[\overset{i}{\underset{k=0}{\sum }}P_{k}\sin\left(\frac{2\pi x_{k}}{v/f}\right)\right]}^{2}},$$
where *f* is the excitation frequency, *v* is the train speed, *N* is the number of train axles, and *x_k_* is the distance between the first axle and the *k*-th axle.

The three main peaks indicated in the figure for the speeds of 150 km/h and 180 km/h are representative of the parametric excitation caused by the passage of the regular groups of axles of the train. As an example, it should be noted that the value of 1.98 Hz refers to the frequency associated with the passage of groups of axles with a regular distance (*l*) of 25.4 m, considering a speed (*v*) of 180 km/h, i.e., *f* = *v* /*λ* = 180/3.6/25.4 = 1.98 Hz.

### 3.2. Numerical Modeling

The numerical model of the BBN vehicle was developed in the ANSYS software version 12.0 [46], consisting of a three-dimensional finite element model including the carbody, bogies, and passenger-seat system. The use of a numerical model based on the finite element method allowed us to consider the influence of the deformability of the various vehicle components. Figure 9 shows a perspective and a cross-section of the model.

The carbody was modeled using shell finite elements, while the bogies were modeled using beam elements, except for the suspensions, which were modeled using spring–dashpot assemblies. Additionally, the passenger-seat systems were modeled using a simplified approach through single-degree-of-freedom systems formed by a mass over a spring–dashpot assembly. The masses of the equipment located in the subgrade of the carbody and in the bogies were taken into account by mass elements. The structure was discretized with 1082 shell elements, 1029 beam elements, and 148 spring–dashpot assemblies. The total number of nodes was 1902, corresponding to 10,704 degrees of freedom.

The numerical model of BBN vehicle was calibrated using an iterative methodology based on a genetic algorithm and resorting on experimental modal parameters, as described in [18]. The experimental campaign carried out on the vehicle involved dedicated dynamic tests of the carbody and bogie. These tests allowed estimating the frequencies and configurations of 13 vibration modes, associated with rigid-body and structural movements of the vehicle and some of its components. Additionally, a dynamic characterization test of the seat-passenger system was also performed [18]. A previous study [18] also detailed the results of the sensitivity analysis and optimization; therefore, for simplicity and to avoid repetition of information, these aspects are herein restricted to essential points. Table 2 presents the values of the main geometric and mechanical parameters of the numerical model of the train. Most of the parameter values refer to the optimal values obtained from the calibration process. The stiffness of the wheel–rail contact was calculated using the Hertz model.

Figure 10 shows the comparison of the vertical modal configurations of the complete vehicle obtained experimentally and numerically. The modal configurations are associated with rigid body modes (1, 2, and 3) and with torsional (4) and bending (5) structural modes of the carbody. In these modes, the bogie movements are negligible. In the same figure, the values of the numerical (f_num_) and experimental (f_exp_) natural frequencies and the corresponding MAC values are also indicated. The results showed a very good agreement between numerical and experimental modal responses. The average error of the natural frequencies was 2.90%, and the average MAC value was equal to 0.937.

## 4. Analysis of the Train–Bridge Dynamic Interaction

The analysis of the bridge–train system was performed by means of an uncoupled methodology that considers the two subsystems, bridge and train, modeled as two independent structures [50]. The points of contact between the two subsystems were between the wheels of the train and the rails. The coupling relation was defined through a unidirectional spring element, in the vertical direction, whose stiffness was a function of the contact force and the geometrical properties of the wheel and the rail, according to Hertz theory. The train bridge dynamic interaction was considered only in the vertical direction, and the loss of contact between the wheel and the rail was not allowed.

### 4.1. Formulation

The dynamic equilibrium equations of the bridge and train subsystems were uncoupled and can be written as follows:(2)$$\left[\begin{array}{cc} M_{b} & 0\\ 0 & M_{v} \end{array}\right]\left[\begin{array}{c} {\ddot{u}}_{b}\\ {\ddot{u}}_{v} \end{array}\right]+\left[\begin{array}{cc} C_{b} & 0\\ 0 & C_{v} \end{array}\right]\left[\begin{array}{c} {\dot{u}}_{b}\\ {\dot{u}}_{v} \end{array}\right]+\left[\begin{array}{cc} K_{b} & 0\\ 0 & K_{v} \end{array}\right]\left[\begin{array}{c} u_{b}\\ u_{v} \end{array}\right]=\left[\begin{array}{c} F_{b}\\ F_{v} \end{array}\right],$$
where the indices *b* and *v* are associated with the matrices or vectors referring to the bridge and the train, respectively, $\ddot{u}$, $\dot{u}$, and u are the vectors of accelerations, velocities, and displacements, *M*, *C*, and *K* are the mass, damping, and stiffness matrices, and *F_v_* and *F_b_* the vectors containing the wheel–rail interaction forces. Table 3 outlines the implementation of the iterative methodology.

In this methodology, the analysis of the train–bridge interaction involves the calculation of the bridge subsystem under the action of the forces transmitted by the vehicles, alternately with the calculation of the train subsystem subjected to the displacements of the bridge [50]. The methodology uses an iterative process at each time increment, aiming at the compatibility of the two subsystems in terms of the dynamic interaction force and the displacements at the points of contact. The compatibility of both subsystems is ensured by the application of a convergence criterion. More details about the iterative methodology can be found in [50].

The solution of the dynamic equilibrium equation of the bridge subsystem was performed using the modal superposition method. This method involves solving a set of uncoupled dynamic equilibrium equations, each one corresponding to a mode of vibration of the structure. The dynamic equilibrium equation of the bridge, relative to the vibration mode *n*, assuming the vibration modes normalized with respect to the mass matrix, is given by
(3)$${\ddot{q}}_{n}+2\xi_{n}\omega_{n}{\dot{q}}_{n}+\omega_{n}^{2}q_{n}=F_{bn},$$
where ${\ddot{q}}_{n}$, ${\dot{q}}_{n}$ and $q_{n}$ represent the modal accelerations, velocities, and displacements, respectively, *ξ_n_* and *ω_n_* are the modal damping and the angular natural frequency respectively, and *F_bn_* is the modal force obtained through the following expression:(4)$$F_{bn}={\tilde{\Phi }}^{T}F_{b},$$
where $\tilde{\Phi }$ is the mode shape of mode *n*. Each uncoupled dynamic equilibrium equation is solved using Newmark’s method.

The displacements at each point of contact ($u_{bc}$) are calculated from the sum of the displacements of the different modes of vibration with the irregularities of the track (*r*), by applying the following expression:(5)$$u_{bc}=\overset{N}{\underset{j=1}{\sum }}{\tilde{\Phi }}_{j}q_{j}+r.$$

The solution of the dynamic equilibrium equation of the train subsystem was also performed by means of the Newmark method, considering the sum of the track roughness and bridge deformation at the respective wheel locations as base excitation.

The forces at the points of contact result from applying the following expression:(6)$$F_{v}=M_{v}^{pf}{\ddot{u}}_{v}+M_{v}^{pp}{\ddot{u}}_{v}+C_{v}^{pf}{\dot{u}}_{v}+C_{v}^{pp}{\dot{u}}_{v}+K_{v}^{pf}u_{v}+K_{v}^{pp}u_{v},$$
where the mass, stiffness, and damping matrices were divided into submatrices, in which *f* represents the free degrees of freedom, and *p* represents the prescribed degrees of freedom. The vectors related to the accelerations (${\ddot{u}}_{v}$), velocities (${\dot{u}}_{v}$), and displacements ($u_{v}$), as well as the forces vector (*F_v_*), are reordered according to the free and prescribed degrees of freedom.

### 4.2. Computational Application

The dynamic analyses of the train–bridge system were performed using a dedicated computational application called TBI (‘train–bridge interaction’) developed by the authors in MATLAB [51]. Figure 11 shows the flowchart illustrating the operational layout of the TBI software.

The main structure of the program consists of five routines, involving the input data (Routine 1), the import data from the ANSYS software (Routine 2), the modal analysis of the bridge (Routine 3), the solution of the train–bridge dynamic interaction problem through the iterative method (Routine 4) and the output of results (Routine 5).

The TBI software includes a set of features that allow increasing its computational efficiency, especially in complex problems involving large-scale models of the bridge and vehicle subsystems. The numerical models of the bridge and the train were performed in ANSYS software, from which the relevant information was extracted (mass matrix, stiffness matrix, etc.) and later integrated into the MATLAB platform through a fully autonomous batch mode interface. The modal analysis of the bridge was carried out in the MATLAB platform and involved the calculation of natural frequencies and modes shapes. The performance of this analysis in the MATLAB environment revealed high efficiency since the mass and stiffness matrices of the bridge system were symmetrical, sparse, and generally of large dimensions. Furthermore, the export of modal results from the ANSYS software was very time-consuming.

The use of the modal superposition method in the solution of the bridge subsystem allowed significantly reducing the time of analysis, mainly in problems with a high number of degrees of freedom. The application of this method simply required the modal information of the load path nodes where applied forces existed. Additionally, the dynamic system formed by the railway vehicles could be divided into autonomous subsystems, called blocks, in correspondence with the different vehicles or groups of vehicles that constituted the train. Identical blocks had the same geometric and mechanical characteristics. The definition of blocks allowed significantly reducing the size of the train subsystem matrices.

## 5. Dynamic Test under Traffic Actions

The dynamic test for the passage of railway traffic allowed the evaluation of the dynamic response in terms of displacements, accelerations, and strains at different points of the deck and arches. The dynamic response, in terms of accelerations, inside the BAS vehicle of the Alfa Pendular train was also evaluated. The measurements on the bridge were synchronized with the measurements on the vehicles through GPS systems.

### 5.1. Experimental Layout

In Figure 12, the instrumented points are shown according to the lateral views of the extrados and intrados arches, along with a plan view and cross-section of the deck. The designation of the instrumented points includes the directions of the measurements according to the represented system of axis.

The displacements were measured in the supports of the abutment on the Porto side (D1 and D2) at the main girder and at the deck slab (D3 and D4) in the section between 1/3 and 1/4 span. The accelerations were measured at the main girder of the deck in the sections between 1/3 and 1/4 span (A1) and midspan (A2). Displacements and accelerations were measured in the vertical direction.

The strains were measured at the start of the extrados arch (S5) and in the P3 hangers (S1 and S2). The strains on the lower face of the main girder (S3) and on the extrados rail (S4) were also measured in the section between 1/3 and 1/4 span. The position of the strain gauges installed on the arches was limited in height for safety reasons due to the proximity of the overhead catenary. All strain gauges were oriented according to the axis alignment of the structural elements.

Data acquisition was performed using the NI cDAQ-9172 system using an NI 9234 module for IEPE type accelerometers, an NI 9239 analog module for conditioning the LVDTs signals, and an NI 9237 strain gauge module. Time records were acquired with a sampling frequency of 2000 Hz, posteriorly decimated to a frequency equal to 200 Hz.

Figure 13 shows some details of the positioning of the displacement transducers, on the deck and on the supports, and of an accelerometer located on the main girder of the deck.

Each displacement transducer was positioned by means of a magnetic base that was supported on a metallic structure fixed to the ground, in the case of the deck (Figure 13a), or resting on the abutment, in the case of the supports (Figure 13b). To measure the displacements of the slab and of the deck’s main girder, two LVDTs, RDP model ACT1000A, were used, with a measurement range equal to ±25 mm, while on the supports two LVDTs, RDP model DCTH100AG, were used, with a measuring range equal to ±2.5 mm. Accelerations were measured using piezoelectric accelerometers, PCB model 393A03, connected to the main girder of the deck through glued metallic plates (Figure 13c).

Figure 14 illustrates the strain gauges installed on hanger P3, with the respective protection system, as well as the strain gauge installed on the rail web. The measurement of the strains of the arch elements was performed by means of electrical resistance strain gauges mounted on a 1/4 Wheatstone bridge scheme with three wires. The rail strain gauge was installed at the neutral axis of the section and in the direction of the longitudinal axis of the track. This sensor allows evaluating the axial strains generated in the rail due to the deck–track composite effect and, thus, characterizing the transmission mechanism of the shear stresses between these two elements.

The measurement of the accelerations in the train was carried out inside the BAS vehicle, in the position identified in Figure 15. The accelerations were measured using a piezoelectric accelerometer, PCB model 393A03, located in the seat base (point A) and positioned through metallic angles fixed with magnetic discs to the seat frame.

### 5.2. Dynamic Responses

This section presents the dynamic responses measured on the bridge for the passage of the Alfa Pendular train. The traffic speeds of the Alfa Pendular train, on the section of the line where the bridge is located, are usually in a range between 175 km/h and 185 km/h. Lower speeds were occasionally recorded, close to 150 km/h, during maintenance works on the track near the bridge.

Figure 16 shows the time records of the displacements of the deck, on the supports (Figure 16a) and on the section between 1/3 and 1/4 span (Figure 16b), of the strains of the hangers (Figure 16c) and of the rail and deck (Figure 16d), for the passage of the Alfa Pendular train at a speed of 180 km/h (AP180).

The analysis of the graphs related to the displacements of the supports shows that the maximum displacement measured in the intrados support (~0.17 mm) was slightly higher than that measured in the extrados support (~0.14 mm) because the deck was not symmetrical in relation to the track axis. Regarding the displacements of the deck, the displacements at the midspan of the slab were greater than those observed on the main girder, with maximum values equal to 2.77 mm and 2.47 mm, respectively. Both responses were dominated by the frequency of 1.98 Hz associated with the passage of the groups of axes with regular distances of 25.4 m. The observation of the records also shows the occurrence of upward displacements on the deck, a fact that is characteristic of structures with an arched structural scheme, in which the application of loads in one of the two half-spans conducts to an anti-symmetrical deformed configuration.

The strain records measured at points S1 and S2 of the P3 hangers (extrados and intrados) show great similarity despite belonging to different arches. In the hangers, the tensile strains reached values of up to 55 μm/m. Regarding the strains measured on the rail and on the deck, the alternation of tensile and compressive stresses between these elements was clearly visible. The maximum and minimum values of the strains were approximately equal to +20 μm/m and −12 μm/m, respectively. This result is very relevant for the characterization of the composite effect between the deck and the track.

Figure 17 shows a comparison between the acceleration records of point A1 on the deck, obtained for the passage of the Alfa Pendular train at speeds of 150 km/h (AP150) and 180 km/h (AP180). The graphs of the normalized average auto-spectrums and the dynamic signature of the train are also presented, in correspondence with the time records, including the identification, in a dashed line, of the values of the train’s parametric excitation frequencies. The acceleration records were filtered by applying a type II Chebyshev low-pass filter with a cutoff frequency equal to 30 Hz.

The comparison of the time records and the auto-spectra referring to the speeds of 150 km/h and 180 km/h allows concluding that the dynamic response of the bridge is greatly influenced by the operating speed of the train. The maximum acceleration recorded for a speed of 180 km/h, equal to 3.1 m/s^2^, was considerably higher than the maximum level of acceleration recorded for a speed of 150 km/h, equal to 0.5 m/s^2^.

For a speed of 150 km/h, the acceleration response was dominated by the quasi-static effect of the axle loads, as can be seen by the importance of the peak with a frequency close to 1.63 Hz, and above all by the anti-symmetrical bending mode (mode 2), with a frequency equal to 4.37 Hz. The relevant contribution of this mode is because its frequency was very close to one of the frequencies associated with the passage of the groups of axles of the train, situated at 4.79 Hz, as shown in the figure, in such a way that the peaks of both frequencies were practically coupled.

For the speed of 180 km/h, it was verified that the excitation frequency, with a value equal to 5.78 Hz, was close to the frequency of the symmetrical bending mode of the deck (mode 3), with a value of about 6 Hz; hence, this mode assumed a greater contribution in the response compared to the participation of mode 2.

For both speeds, the frequency spectra show several peaks for frequencies higher than 9 Hz, which were probably related to the bridge’s higher order modes (e.g., modes 5, 7, 8, 10) activated due to the influence of the track irregularities.

### 5.3. Damping Ratios

The modal damping ratios were estimated on the basis of the logarithmic decrement method and on the analysis of the records of accelerations in free vibration after a train passage [52].

The logarithmic decrement method involves applying a digital bandpass filter to the acceleration record, around the frequency of the mode for which the damping ratio is to be estimated, followed by fitting an exponential function to the maximums of the filtered record:(7)$$a=Ce^{-\xi \omega t},$$
where *ω* is the angular natural frequency, *C* is a constant, and *ξ* is the damping ratio.

Figure 18 illustrates the application of this method in estimating the damping ratios of modes 2 and 3, considering 15 cycles of the initial zone or an intermediate zone of the free vibration response from location A1 of the deck to the passage of the train AP150. The record was filtered using bandpass filters of the Chebyshev II type, with band attenuation equal to 45 dB and bandwidth equal to 3 Hz.

Table 4 shows the values of the damping ratios of modes 2 and 3 obtained on the basis of the bridge acceleration records for the passage of trains AP150 and AP180. The remaining bridge vibration modes had reduced amplitudes, which made it difficult to estimate the respective damping ratios.

The table shows that the values of the damping ratios calculated considering the initial zone of the response in free vibration were slightly higher than those calculated considering an intermediate zone, i.e., for vibration mode 2. This result confirms the damping growth trend with the increase in the vibration level. The values of the damping ratios also show a reduced variability as a function of the train speed. It was also verified that the values of the modal damping ratios were higher than that specified in EN 1991-2 [53] for bridges with a mixed steel–concrete deck and spans greater than 20 m, i.e., 0.50%.

## 6. Experimental Validation of the Dynamic Model of the Train–Bridge System

The experimental validation of the numerical model consisted of comparing the dynamic responses obtained from the train–bridge interaction model, including the track irregularities and the responses obtained experimentally, through the dynamic test under traffic actions.

The numerical analyses were performed in the TBI software considering the contribution of 85 vibration modes to the bridge response, with frequencies between 2.34 Hz and 30 Hz. The analysis time step was equal to 0.001 s, and the free vibration period was considered equal to 3 s. The computational time spent in the calculation of the train–bridge dynamic analyses was approximately 22 h 15 m and 20 h 45 m, for train speeds equal to 150 km/h and 180 km/h, respectively, using a computer with two processors Intel^®^ XEON E5430 at 2.67 GHz and 28 Gb RAM.

For the experimentally identified vibration modes, the damping coefficients adopted were equal to the average values of the coefficients obtained through the ambient vibration test, according to [47]. For the remaining modes, damping coefficients equal to 0.5% were considered. The effect of track flexibility associated with the inclusion of the vertical vibration modes of the track was not considered. Experimental responses were filtered by applying a type II Chebyshev low-pass filter with a cutoff frequency equal to 30 Hz.

### 6.1. Displacements

Figure 19 shows a comparison of the time records and the correspondent auto-spectra of the vertical displacement of the support (location D1) and the deck (location D3), obtained experimentally and numerically for the passage of the Alfa Pendular train at a speed of 180 km/h.

The figures show a very good agreement between the experimental and numerical records. This agreement is visible for both the forced vibration and the free vibration movements. Regarding the frequency content, the amplitudes of the response peaks associated with the loading (1.98 Hz, see Figure 8) and mode 3 (5.91 Hz) were matched with the amplitudes of the experimental peaks. The peak associated with mode 12, with a frequency greater than 20 Hz, and whose shape represented the bending of the deck with movements of the supports, was also quite close to the peak identified experimentally. The contribution of mode 9 (13.48 Hz), a deck torsion mode that appeared in the experimental response, was not identified in the numerical frequency response. This effect was probably associated with some slight eccentricity in the application of the train loads, due to some unbalance between the left- and right-handed wheels, which caused the torsion of the deck. This effect was not captured in the numerical model where balanced axle loads were considered.

### 6.2. Accelerations

Figure 20 shows a comparison of the time records and the respective auto-spectra of the vertical acceleration on the deck (location A1), obtained experimentally and numerically, for the passage of the Alfa Pendular train at a speed of 150 km/h. Numerical results were obtained considering a train–bridge interaction methodology, without and with the inclusion of the irregularities, as presented in Figure 20a,b, respectively.

Generally, the numerical records were very similar to the experimental record. However, a more detailed analysis to the response in the frequency domain shows that the contributions of frequencies above 9 Hz, which appeared in the auto spectrum of the experimental record, did not appear in the numerical record without the contribution of track irregularities. Moreover, in case of the numerical record without the contribution of track irregularities, two main frequencies were present in the dynamic response: a frequency related to passage of the regular groups of axles of the train (1.63 Hz, see Figure 8) and the frequency of mode 2 (4.37 Hz).

The inclusion of track irregularities was decisive for increasing the correlation between numerical and experimental records. For the numerical record considering the effect of track irregularities, two aspects became clearly visible: (1) the existence of peaks at frequencies between 9 Hz and 25 Hz and with amplitudes close to the amplitudes of the experimental peaks; (2) an increase in the amplitude of the peak related to the quasi-static effect of the traffic loads at a frequency of 1.63 Hz.

The participation of the higher-order modes of the bridge, mode 5 (9.76 Hz), mode 7 (11.30 Hz), mode 8 (13.76 Hz), and mode 10 (15.80 Hz), was related to the influence of the track irregularities with shorter wavelengths, particularly in the range between 3 m and 5 m. The peak close to 25 Hz was associated with a higher-order mode of the bridge not experimentally identified under ambient actions and not used in the numerical model calibration [47]. This specific frequency peak was probably excited due to the contribution of the vehicle dynamics, i.e., the vibrations of the bogies, which can be excited by the short-wave irregularities. Despite the track irregularities in the numerical model being limited to 3 m, an even shorter wavelength would be required to mobilize a frequency close to 25 Hz, and these irregularities were able to amplify the response in this frequency domain.

Regarding the increase of the peak at a frequency of 1.63 Hz, the effect of the track irregularities was also very important, since the influence of larger wavelengths, in this case close to 25 m (see Figure 6), would lead to an amplification of the dynamic response.

### 6.3. Strains

Figure 21 shows a comparison of the time records and respective auto spectra of the strains at the start of the arch (location S5), at the hanger P3 (location S1), and at the rail (location S4), obtained experimentally and numerically, for the passage of the Alfa Pendular train at a speed of 180 km/h.

Generally, the experimental and numerical strain records showed a good agreement and were dominated by the contribution of the frequency associated with the passage of the groups of axles of the train with a distance equal to 25.4 m, equal to 1.98 Hz.

Regarding the strains at the start of the arch and hangers, there were visible differences in the contributions of the higher frequencies, i.e., for mode 3 (6.02 Hz), as can be seen by the difference in the amplitudes of the experimental and numerical auto spectra. This vibration mode seemed to have an increasing participation as the train crosses the bridge, as stated in the experimental results for the arches, and this effect was not properly captured by the numerical model. This phenomenon may have been due to changes in the stress state of the metallic elements during the passage of traffic, particularly those belonging to the arches, which may have caused slight variations of the natural frequencies. Probably for the same reason, the contribution of mode 8 (13.76 Hz), a global mode that involved the transversal bending of the arches, was not adequately reproduced in the numerical response at the location S5 of the arch.

### 6.4. Accelerations in BAS Vehicle

Figure 22 shows a comparison of the time records and the respective auto spectra of the vertical acceleration of point A of the seat base in the BAS vehicle of Alfa Pendular train, obtained experimentally and numerically, for crossing the São Lourenço bridge at a speed of 180 km/h.

The experimental and numerical records showed a reasonable agreement and had maximum acceleration values of the same order of magnitude (close to 0.5 m/s^2^). The seat base acceleration record was predominantly influenced by the frequency associated with the rigid body movements of the carbody, namely, the bouncing movement, with a frequency equal to 1.33 Hz (mode 2 in Figure 10).

## 7. Conclusions

This article describes the experimental validation of a train–track–bridge dynamic interaction model of a bowstring arch bridge. For this purpose, highly complex and calibrated 3D numerical models of both train and bridge subsystems were developed. Dynamic analyses of the train–bridge system were performed on a dedicated computational tool, the TBI software, which considered their dynamic interaction in the vertical direction by means of an uncoupled methodology. The track irregularity profiles obtained from measurements performed by a track inspection vehicle were also considered.

The results of the experimental test under railway traffic allowed stating the occurrence of upward displacements on the deck, a fact characteristic of structures with an arched structural behavior. Additionally, the longitudinal strains measured on the rail and deck showed an alternation of tensile/compressive stresses between these two elements, which was relevant to characterize the track–deck composite effect. The accelerations measured for two distinct train speeds proved that the dynamic response of the bridge was influenced by the train’s operating speed. In addition to the influence of the quasi-static effect of the axle loads, the acceleration responses were dependent on the anti-symmetrical and symmetrical bending modes of the bridge. Furthermore, the values of the damping ratios derived from the application of the decrement logarithm method confirmed the damping growth trend with the increase in the vibration level.

The validation of the train–bridge system involved the comparison of the dynamic responses of the bridge and train numerically obtained with the responses derived from the experimental test under traffic. In general, a very good correlation was obtained between numerical and experimental results in terms of accelerations, displacements, and strains on the bridge elements. The contributions derived from the parametric excitation of the train, the global/local dynamic behavior of the bridge, and the excitation derived from the track irregularities were decisive to accurately reproduce the complex behavior of the train–track–bridge system. For bridge deck accelerations, the contribute of track irregularities were decisive to enhance the matching between numerical and experimental records, particularly those with shorter wavelengths, in the range between 3 m and 5 m, as well as larger wavelengths, close to 25 m. Regarding the strains at the arch elements, there were still some visible differences between numerical and experimental records, due to the contributions of higher-order modes, which seemed to have an increasing participation as the train crossed the bridge. This effect was probably caused by the changes in the stress state of the metallic elements during the passage of traffic, which may have caused slight variations of the natural frequencies not properly captured by the numerical model. In what concerns the acceleration inside the train, a reasonable agreement between numerical and experimental records was achieved, with maximum values of the same order of magnitude. It was also shown that the seat base acceleration record was predominantly influenced by the bouncing movement of the carbody.

Thus, as the main outcome of the present study, it should be highlighted the development of a highly complex dynamic model of the train–track–bridge system, able to reliable capture the global dynamic behavior of the train and bridge subsystems under real operational scenarios.

Despite the efficiency of the validated model in predicting the train–track–bridge global dynamic responses, it should be pointed out some of its limitations, especially in capturing (i) the local dynamic behavior of the bridge, particularly on specific critical details of the arches, which would require the adoption of a sub-modeling approach, (ii) the local dynamic behavior of the track components, which is greatly influenced by higher-frequency modal contributions not considered in this study, and (iii) the eventual nonlinear incursions of the bridge subsystem, i.e., at the support bearings, at the track–deck interface, and at the semirigid connections between arch elements.

Lastly, in ongoing and future work, using the validated train–track–bridge dynamic model, the authors intend to develop an advanced methodology for structural damage identification based on artificial intelligence [54,55].

## Figures and Tables

**Figure 1 sensors-23-00171-f001:**
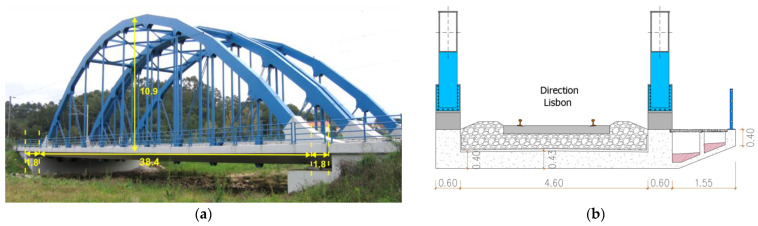
São Lourenço bridge: (**a**) lateral view; (**b**) half-deck cross-section.

**Figure 2 sensors-23-00171-f002:**
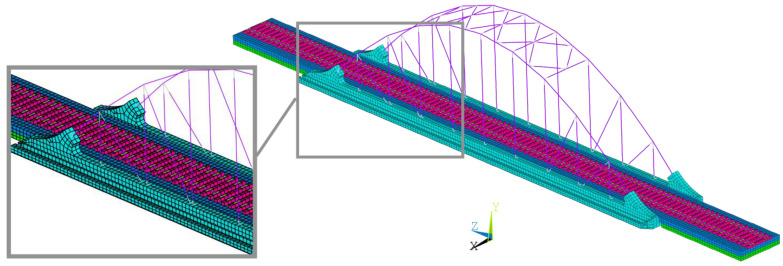
The 3D model of the São Lourenço bridge including the track.

**Figure 3 sensors-23-00171-f003:**
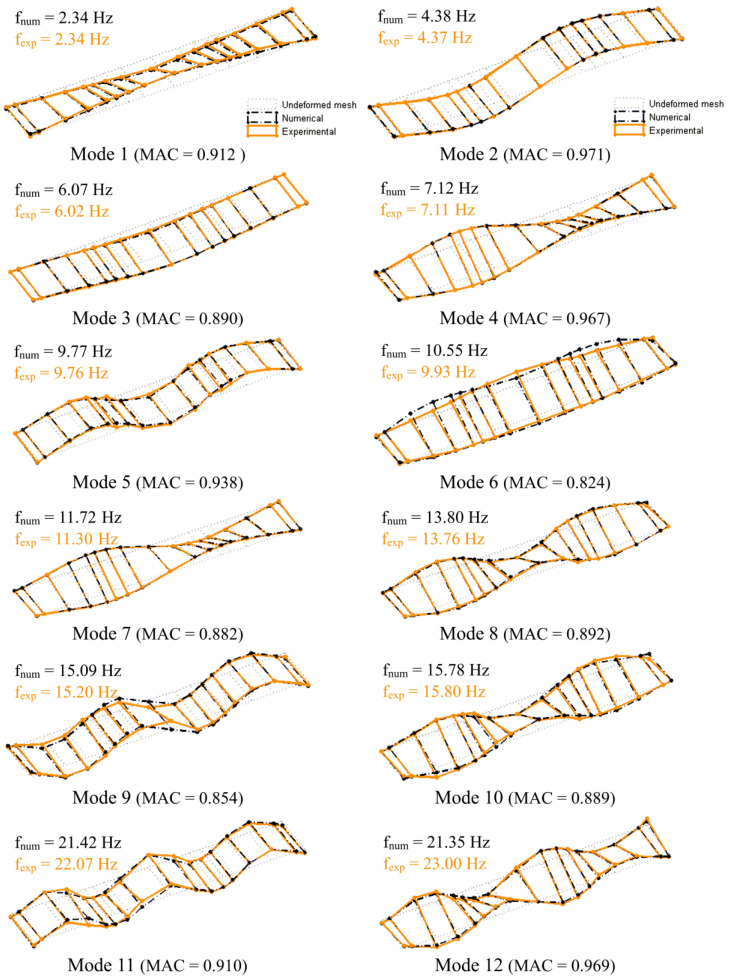
Comparison of the global vibration modes of the bridge obtained experimentally and numerically.

**Figure 4 sensors-23-00171-f004:**
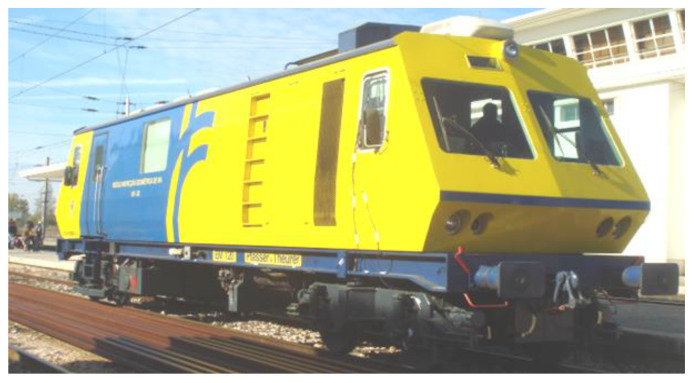
Track inspection vehicle EM 120 from IP.

**Figure 5 sensors-23-00171-f005:**
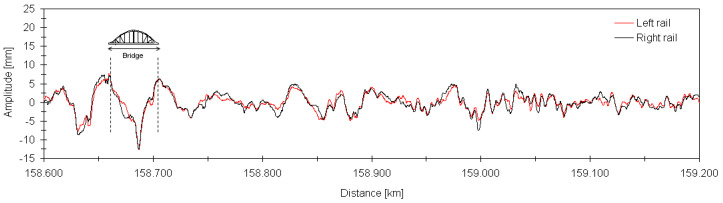
Longitudinal leveling of the left and right rails.

**Figure 6 sensors-23-00171-f006:**
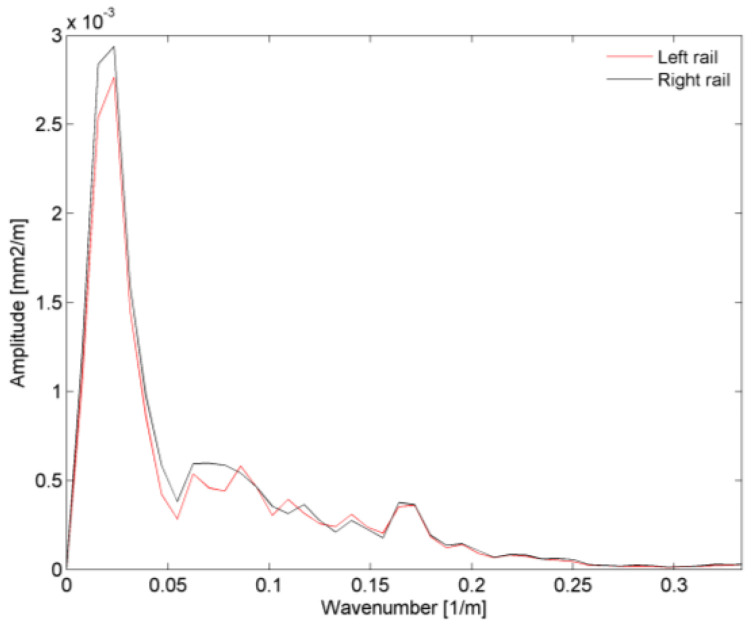
Amplitude of the power spectra of the irregularities, as a function of the wave number, on the left and right rails.

**Figure 7 sensors-23-00171-f007:**
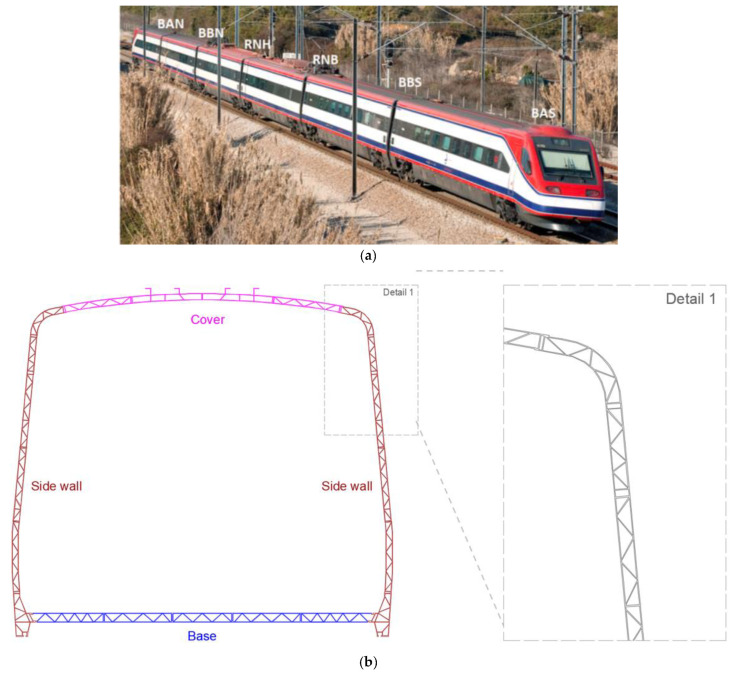
Alfa pendular train: (**a**) perspective view; (**b**) cross-section of the carbody and structural detail.

**Figure 8 sensors-23-00171-f008:**
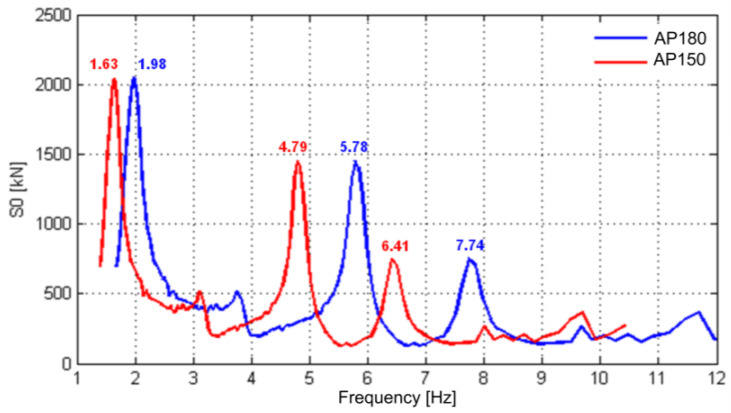
Dynamic signature of the Alfa Pendular train for speeds of 150 and 180 km/h.

**Figure 9 sensors-23-00171-f009:**
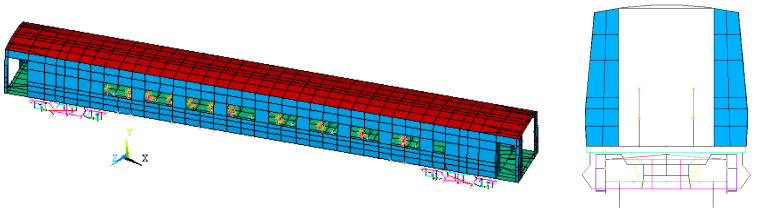
Numerical model of BBN vehicle.

**Figure 10 sensors-23-00171-f010:**
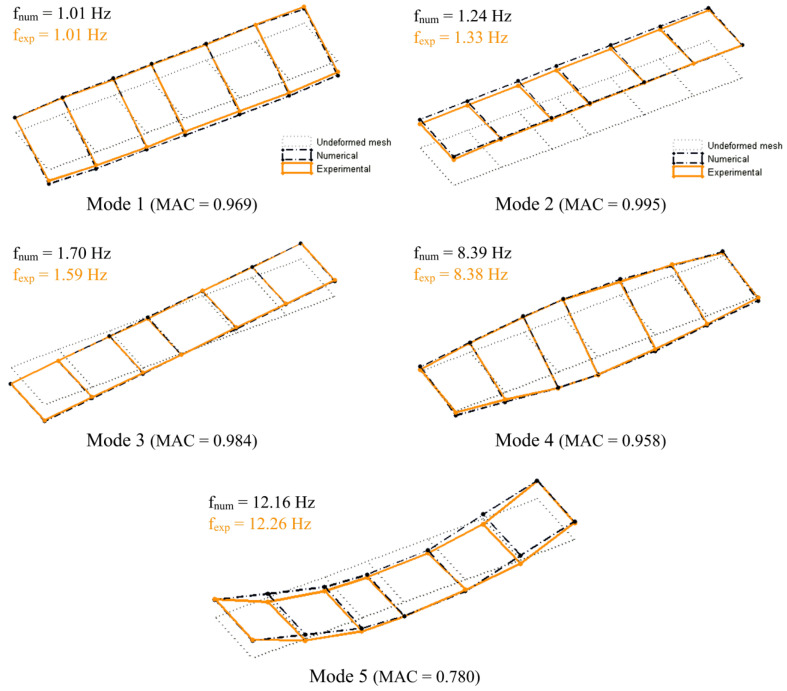
Comparison of the vibration modes of the BBN vehicle obtained experimentally and numerically.

**Figure 11 sensors-23-00171-f011:**
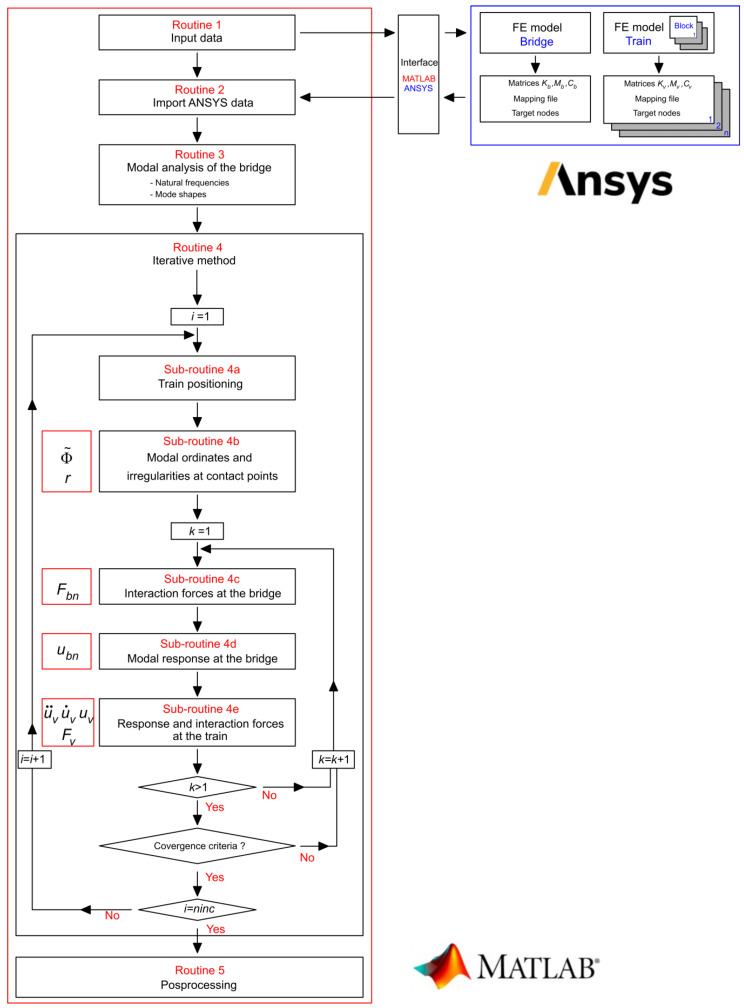
Flowchart of TBI software.

**Figure 12 sensors-23-00171-f012:**
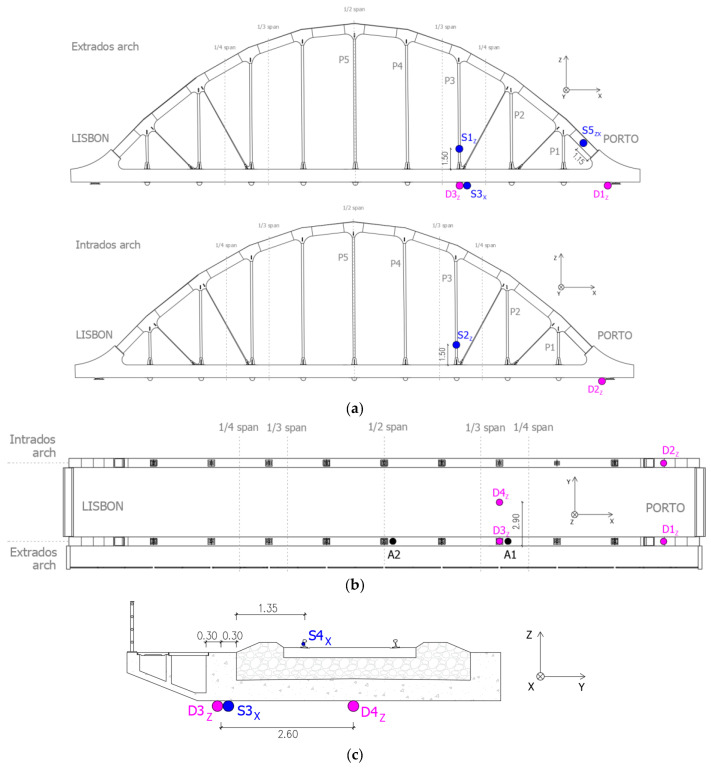
Measuring points in the dynamic test under railway traffic: (**a**) lateral views of the arches (extrados and intrados); (**b**) plan view of the deck; (**c**) deck cross-section (between 1/3 and 1/4 span).

**Figure 13 sensors-23-00171-f013:**
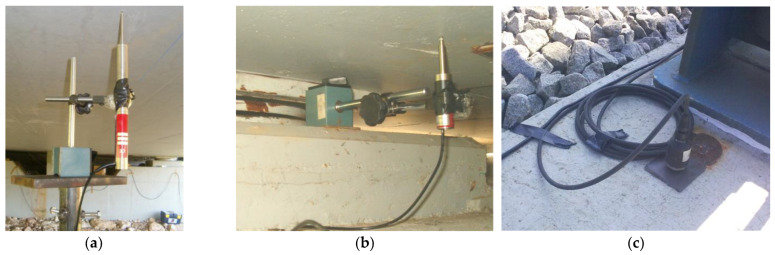
Measurement of displacements and accelerations: (**a**) LVDT on the deck slab; (**b**) LVDT on the support; (**c**) accelerometer on the main girder.

**Figure 14 sensors-23-00171-f014:**
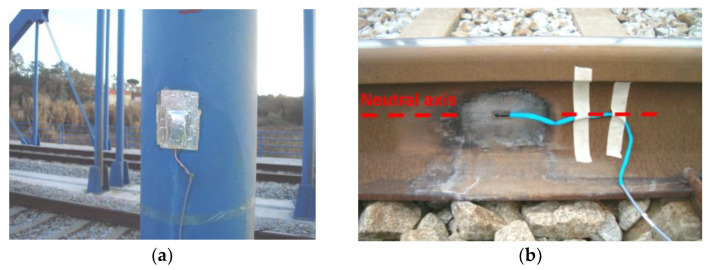
Strain measurement: (**a**) hanger P3; (**b**) rail.

**Figure 15 sensors-23-00171-f015:**
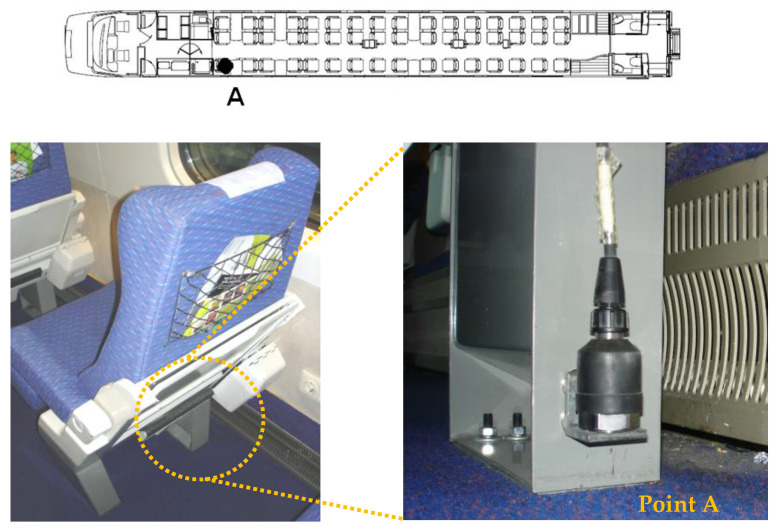
Measurement of acceleration inside the BAS vehicle of Alfa Pendular train.

**Figure 16 sensors-23-00171-f016:**
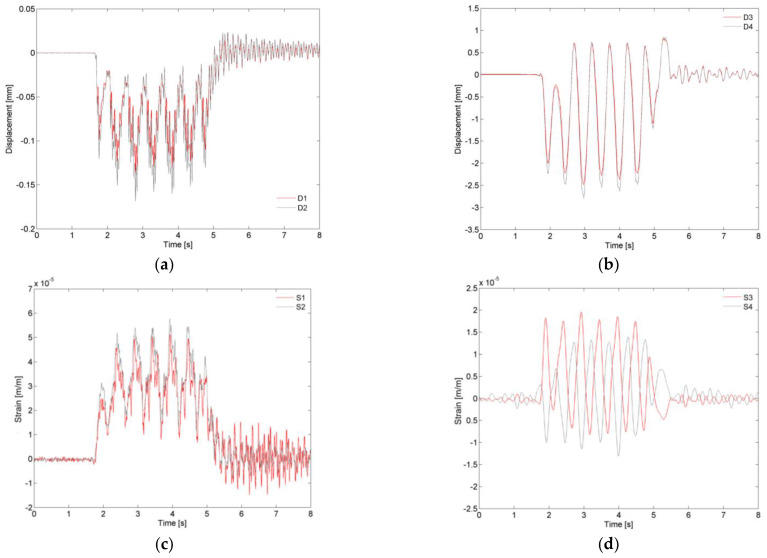
Experimental records of the passage of the Alfa Pendular train at a speed of 180 km/h: (**a**) displacement of the supports; (**b**) displacement of the main girder and deck slab; (**c**) strain of the hangers P3; (**d**) strain of the main girder and rail.

**Figure 17 sensors-23-00171-f017:**
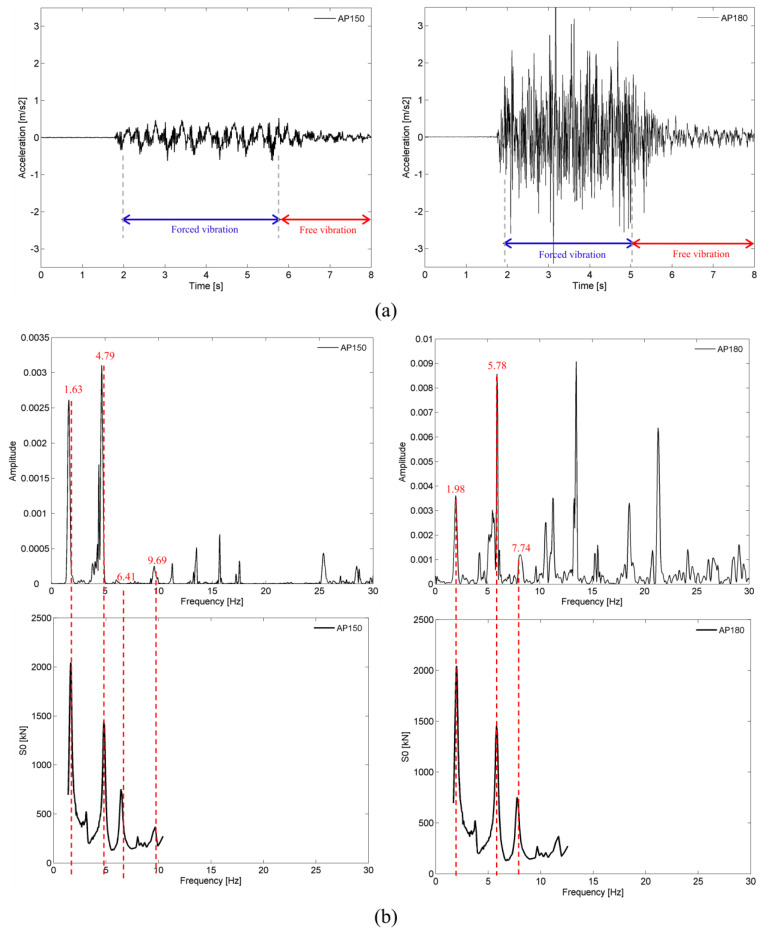
Vertical accelerations of the deck at point A1 for the passage of the Alfa Pendular train at speeds of 150 km/h (AP150) and 180 km/h (AP180): (**a**) time records; (**b**) auto-spectrum and dynamic signatures of the train.

**Figure 18 sensors-23-00171-f018:**
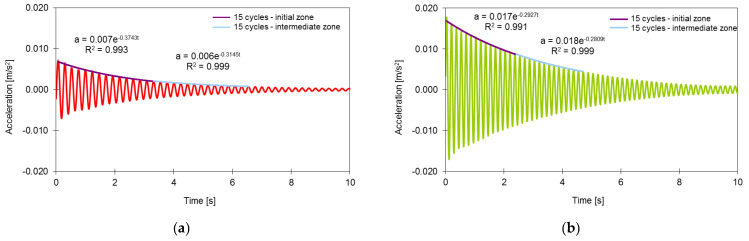
Application of the logarithmic decrement method to the AP150 train acceleration record, in estimating the damping ratios: (**a**) mode 2; (**b**) mode 3.

**Figure 19 sensors-23-00171-f019:**
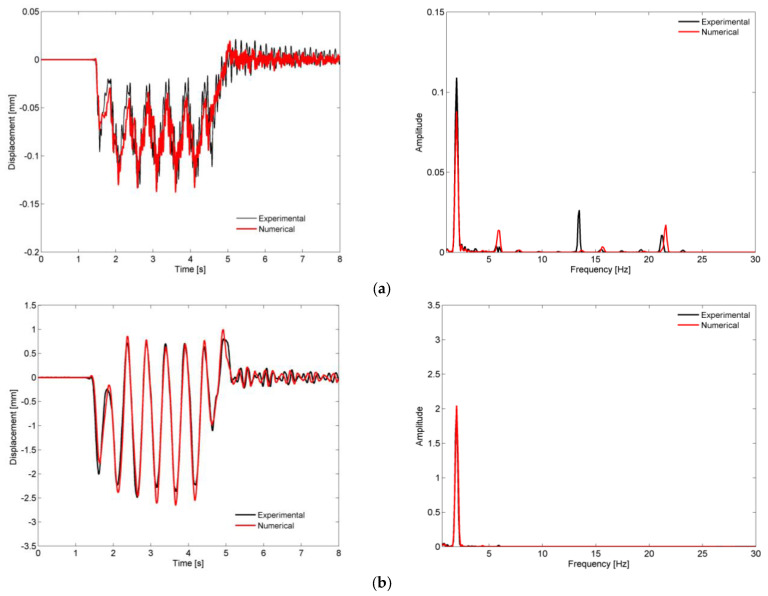
Comparison of displacement records and respective auto-spectra for the passage of Alfa Pendular train at a speed of 180 km/h, obtained experimentally and numerically, at locations (**a**) D1 and (**b**) D3.

**Figure 20 sensors-23-00171-f020:**
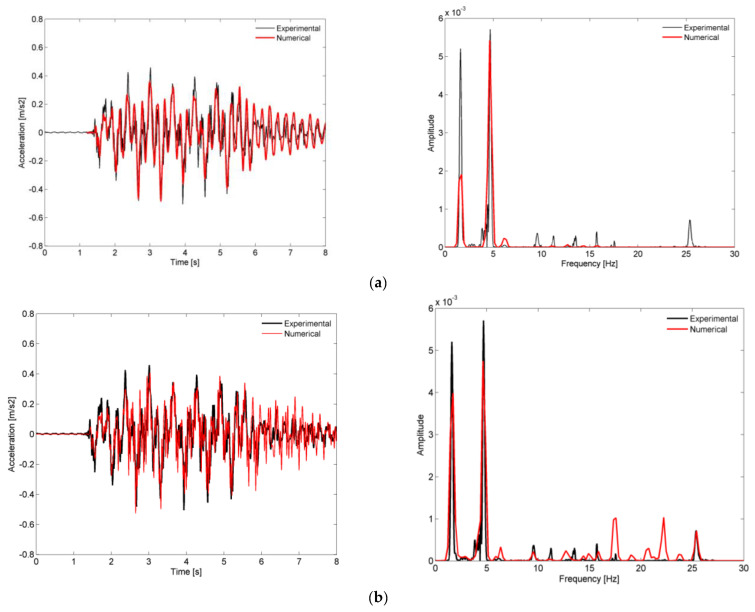
Comparison of the records of the vertical acceleration on the deck (location A1) and the respective auto-spectra, obtained experimentally and numerically, for the passage of the Alfa Pendular train at a speed of 150 km/h: (**a**) without irregularities; (**b**) with irregularities.

**Figure 21 sensors-23-00171-f021:**
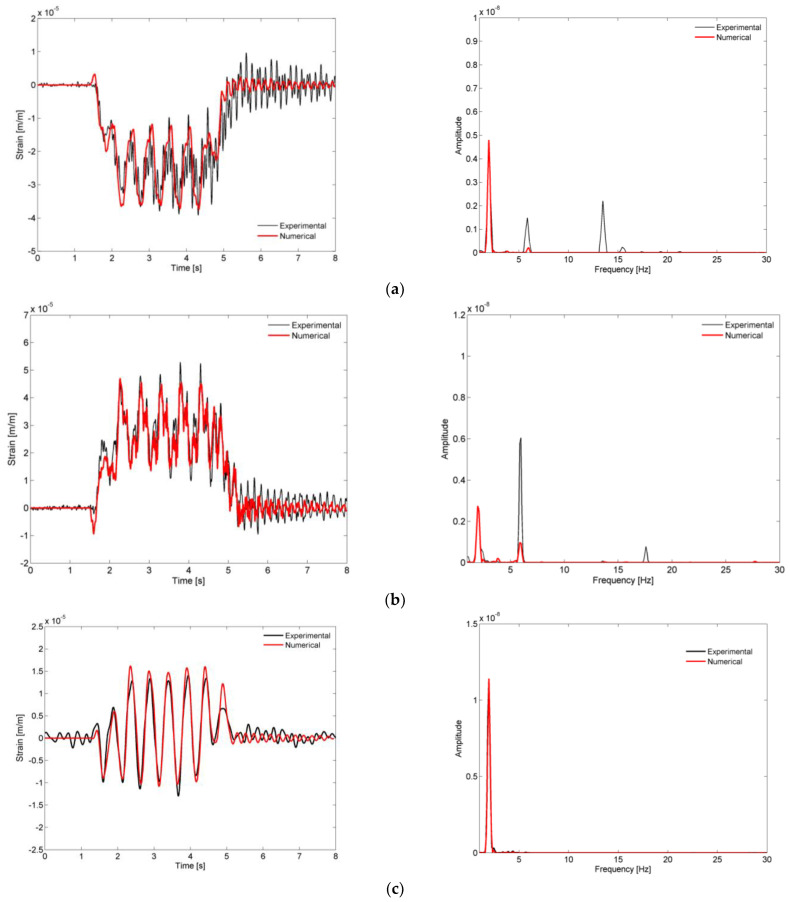
Comparison of strains records and respective auto-spectra for the passage of the Alfa Pendular train at a speed of 180 km/h, obtained experimentally and numerically, from locations (**a**) S5, (**b**) S1, and (**c**) S4.

**Figure 22 sensors-23-00171-f022:**
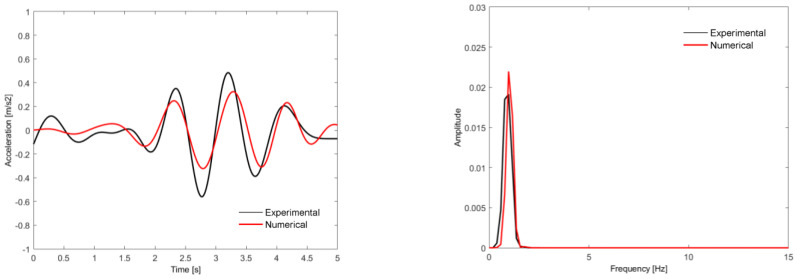
Comparison of the time records and respective auto spectra of the vertical acceleration of point A of the seat base from vehicle BAS, obtained experimentally and numerically, for the passage over São Lourenço bridge at a speed of 180 km/h.

**Table 1 sensors-23-00171-t001:** Characterization of the main parameters of the numerical model of São Lourenço bridge.

Parameter	Designation	Value	Unit
E_c_	Elasticity modulus of concrete	44.2	GPa
ρ_c_	Density of concrete	2423	kg/m^3^
E_bal_	Elasticity modulus of ballast	148.1	MPa
ρ_bal_	Density of ballast	1743	kg/m^3^
E_s_	Elasticity modulus of steel	202.5	GPa
ρ_s_	Density of steel	7850	kg/m^3^
K_v_	Vertical stiffness of pot bearings	5210	MN/m
K_palm_	Vertical stiffness of rail pads	400	MN/m
A_c_/I_c_	Area/inertia of rail UIC54	69.3/2346	cm^2^/cm^4^
A_arc_/I_arc_	Area/inertia of arch (current section)	374.4/275,700	cm^2^/cm^4^
A_P_/I_P_	Area/inertia of hangers	49.7/1560	cm^2^/cm^4^
A_diag_/I_diag_	Area/inertia of diagonals	39.3/61.4	cm^2^/cm^4^

**Table 2 sensors-23-00171-t002:** Characterization of the main numerical model parameters of the Alfa Pendular train.

Parameter	Designation		Value	Unit
K_S_	Stiffness of secondary suspensions		367.4 (front) 343.1 (rear)	kN/m
c_S_	Damping of secondary suspensions		35	kN·m/s
E_alum_	Elasticity modulus of aluminum		70	GPa
ρ _alum_	Density of aluminum		2700	kg/m^3^
RMI	Corrective factor for the inertia of the carbody panels	Base	83.4	-
		Lateral walls	16.1	-
		Roof	386	-
e	Equivalent thickness of carbody panels	Base	10.2	mm
		Lateral walls	10.3	mm
		Roof	8.8	mm
∆M	Additional mass on carbody panels	Base	58	%
		Lateral walls	20	%
		Roof	11	%
K_P_	Stiffness of primary suspensions		826.4	kN/m
c_P_	Damping of primary suspensions		16.7	kN·m/s
K_RC_	Wheel-rail contact stiffness		1.5674	×10^9^ N/m

**Table 3 sensors-23-00171-t003:** Iterative methodology for solving the train-bridge dynamic interaction problem.

Scheme	Subsystem	Problem Solver Method	Input	Output
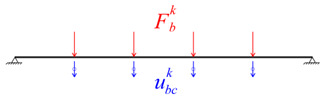	Bridge	Modal Superposition	$F_{b}^{k}=F_{v,sta}+F_{v}^{k-1}$	$u_{bc}^{k}$
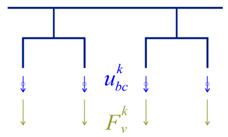	Vehicles	Direct Integration	$u_{bc}^{k}$	$F_{v}^{k}$

**Table 4 sensors-23-00171-t004:** Damping ratios values for modes 2 and 3 depending on the vibration level.

Vibration Mode	Train	Damping Ratios *ξ* [%]	
		Initial Zone	Intermediate Zone
2	AP 150	1.35	1.14
	AP 180	1.34	1.12
3	AP 150	0.77	0.74
	AP 180	0.83	0.80

## Data Availability

Data is unavailable due to privacy restrictions.

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
