# Peer review of "Train–Track–Bridge Dynamic Interaction on a Bowstring-Arch Railway Bridge: Advanced Modeling and Experimental Validation"

_sensors, 2022, doi:10.3390/s23010171_

Round 1

Reviewer 1 Report

In this manuscript, the authors develop a train-track-bridge dynamic interaction model of the São Lourenço bridge and validate the system model through field measurement data very well. The presented work is quite interesting, informative, and useful to readers of this journal. In addition, the manuscript is also very well written and structured. I recommend it for publication and have only minor comment.

- It is not clear how much time the TBI analysis requires in the present study, can the author please give some information about this?

- A fine FE model of the bridge was built by authors. If possible, can the author please comment whether the model can be adopted to investigate fatigue of the railway bridge which is the topic of previous researches (such as https://doi.org/10.3390/app10093261 and https://doi.org/10.1061/(ASCE)BE.1943-5592.0001144).

- Please check the symbols for strain measuring point, it seems they are not consistent in the context.

Reviewer 2 Report

The authors describe the validation of a 3D dynamic interaction model of the train-track-bridge system on a bowstring-arch railway bridge based on experimental tests. I think the article is meaningful and can be published after the following questions are revised.

1.      More details of the ANSYS model are required, such as the types of elements used, the specific size of the track structure (sleepers, rail pads and ballast layer), etc.

2.      The text and figures are not clear enough in Figure 12 a and c.

3.      Since bridges and trains are modeled finely, computational efficiency may need to be considered. It is better for the author to record the computational time of some examples and to demonstrate whether this method requires high computer performance.

Reviewer 3 Report

Dear Editor

The authors conduct a study to validate a 3D dynamic interaction model of the train-track-bridge system on a bowstring-arch railway bridge based on experimental tests. The model sub systems are modelled based on finite element model previously calibrated based on experimental data and the authors found the model results to be in a very good correlation.

This study holds good efforts and contributes to the literature. However, there are some issues that the authors should consider in the revision round.

Please consider the following comments.

General Comments:

Limitations of the previous studies should be discussed.

Outcomes and limitations of the present study regarding the validated model calibrations should be discussed. May this model validate out of the limitations?

In last bullet of page 6, please replace “estimations” with “estimates”.

There are some typos. They should be revised. For instance, in page 7, pay attention “length”.

Version of ANSYS should be given.

Please give full form of MAC.

In Table 1, the optimal values based on the calibration process were given. If the values in Tables are not obtained based on the experimental data, the optimization procedure should be discussed. How did the authors search these parameters and conclude the optimal values? However, the authors should discuss the equifinality problem of the parameters.

What is the sensitivity of the parameters given in Table 1? Please address the sensitivity of each parameter value. The changes in parameter values given in Table 1 may affect the results of Figure 3. What about the reliability of the model outputs?

If the authors could add legend in Figure 3, the deformations may be easily detected.

The similar problems may be discussed for the values given in Table 2 and the results presented in Figure 10.

Authors may prefer including a new section to address this issue.

Equations should be given properly and written in equation in word.

Round 2

Reviewer 3 Report

The manuscript can now be accepted for publication.